# Characterization of *GPX* Gene Family in Pepper (*Capsicum annuum* L.) under Abiotic Stress and ABA Treatment

**DOI:** 10.3390/ijms25158343

**Published:** 2024-07-30

**Authors:** Zeyu Zhang, Jing Zhang, Cheng Wang, Youlin Chang, Kangning Han, Yanqiang Gao, Jianming Xie

**Affiliations:** College of Horticulture, Gansu Agricultural University, Lanzhou 730070, China; zhangzey@st.gsau.edu.cn (Z.Z.); zj@gsau.edu.cn (J.Z.); wangcheng@gsau.edu.cn (C.W.); cyl@st.gsau.edu.cn (Y.C.); hkn@st.gsau.edu.cn (K.H.); gyq@st.gsau.edu.cn (Y.G.)

**Keywords:** pepper, glutathione peroxidase, abiotic stress, expression pattern

## Abstract

Plant glutathione peroxidases (GPXs) are important enzymes for removing reactive oxygen species in plant cells and are closely related to the stress resistance of plants. This study identified the *GPX* gene family members of pepper (*Capsicum annuum* L.), “CM333”, at the whole-genome level to clarify their expression patterns and enzyme activity changes under abiotic stress and ABA treatment. The results showed that eight *CaGPX* genes were unevenly distributed across four chromosomes and one scaffold of the pepper genome, and their protein sequences had Cys residues typical of the plant GPX domains. The analysis of collinearity, phylogenetic tree, gene structure, and conserved motifs indicated that the *CaGPX* gene sequence is conserved, structurally similar, and more closely related to the sequence structure of *Arabidopsis*. Meanwhile, many cis elements involved in stress, hormones, development, and light response were found in the promoter region of the *CaGPX* gene. In addition, *CaGPX1*/*4* and *CaGPX6* were basically expressed in all tissues, and their expression levels were significantly upregulated under abiotic stress and ABA treatment. Subcellular localization showed that CaGPX1 and CaGPX4 are localized in chloroplasts. Additionally, the variations in glutathione peroxidase activity (GSH-Px) mostly agreed with the variations in gene expression. In summary, the *CaGPXs* gene may play an important role in the development of peppers and their response to abiotic stress and hormones.

## 1. Introduction

Glutathione peroxidase (GPX; EC 1.11.1.9) is not only an important member of the ascorbate–glutathione cycle (AsA-GSH) but also an important enzyme for clearing ROS in cells [1,2]. It belongs to the nonheme thiol peroxidase family [3], which can use glutathione (GSH) or thioredoxin reductase (TRX) as reducing agents to catalyze the reduction of hydrogen peroxide (H_2_O_2_), organic hydrogen peroxide oxides, and lipid peroxides into water or corresponding alcohols, thereby maintaining the balance of ROS in plants and protecting cells from the toxic effects of high concentrations of ROS [4,5]. People’s understanding of GPX and research on its functional mechanism started in animals; it was discovered in 1957 by extracting mammalian red blood cells for enzyme testing and H_2_O_2_ reaction [6]. Since GPX in mammals uses GSH as an electron donor to reduce peroxides such as H_2_O_2_, the name GPX was derived from this [7]. Studies of GPX in plants started late. Criqui et al. [8] identified the first plant-derived GPX in tobacco, isolated it, and then successively found GPX-homologous genes in different plants. In plants, GPX usually exists in the form of a monomer [9], and almost all eukaryotic genomes contain coding genes of the GPX family, exhibiting high sequence similarity in conserved motifs and domains [10]. Compared with mammalian GPXs, plant GPXs contain cysteine (Cys) in their active sites, and when a large number of peroxides accumulates in the plant, the three conserved Cys residues at the N-terminus are converted into sulphenic acid, which forms an intramolecular disulfide bond with the separated conserved Cys residues [11,12]. The disulfide bond is subsequently reduced by the thioredoxin (Trx) or glutathione (Grx) system; while plant GPXs are usually reduced by Trx [13], most mammalian GPXs contain selenocysteine instead of Cys residues in their catalytic active sites, which can be reduced by Grx and Trx [14].

In recent years, researchers have cloned *GPX* family genes from *Arabidopsis*, *Oryza sativa*, and other plants and verified that changes in GPX gene expression are crucial for plants to respond to biotic and abiotic stresses [15,16,17]. In *Arabidopsis*, a total of eight members of the *AtGPX* family were identified, among which both *AtGPX2* and *AtGPX6* were localized in the cytoplasm and cell membrane [18]. The sensitivity of *AtGPX3* and *AtGPX8* mutant plants to drought and osmotic stress was significantly higher than that of wild-type *Arabidopsis* plants [19], and the overexpression of the *AtGPX5* gene in *Arabidopsis* leads to changes in plant growth and redox status under salt stress [20]. In rice, the *OsGPX* gene family has six members, and *OsGPX1*-*4* is upregulated by H_2_O_2_ induction [21,22]. In addition, studies have shown that the knockout of *OsGPX1* and *OsGPX3* in rice may seriously interfere with rice growth and development [23]. The knockout of *OsGPX5* increases the sensitivity of rice to salt stress and impairs seed germination and plant development [24]. Silencing *OsGPX1* located in the mitochondria blocks photosynthesis and photorespiration in rice under salt stress [25]. A total of 13 *GhGPX* gene family members have been identified in *Gossypium hirsutum*, among which *GhGPX1* was localized in chloroplasts, *GhGPX3* and *GhGPX5*-*8* were localized in the cytoplasm, and *GhGPX13* was upregulated by the induction of high temperatures [26]. In sorghum, there were seven members of the *SbGPX* gene family, among which *SbGPX2*-*5* was upregulated by drought stress and *SbGPX6*-7 was downregulated by drought stress [27]. In addition, the overexpression of the *GPX* gene can improve the tolerance of transgenic plants to abiotic stress, such as *Rhodiola crenulata RcGPX5* overexpressed in *Salvia miltiorrhiza*, *Pinus massoniana PmGPX6* overexpressed in *Arabidopsis*, and wheat *TaGPX* genes *W69* and *W106*, which can enhance the ability of transgenic plants to remove ROS and improve the tolerance of plants to salt, alkali, and drought [28,29,30,31].

Pepper (*Capsicum annuum* L.) is an important cash crop, a vegetable spice crop, and a value-added processing product widely planted worldwide [32,33]. However, during the cultivation process, pepper is highly vulnerable to both biotic and abiotic stress, which can impede its growth and development, ultimately affecting both its yield and quality. GPX is an effective reactive oxygen species scavenger in plants and is crucial for plants’ resistance to stress. However, the *GPX* gene family has not been identified and reported in peppers. Therefore, we utilized bioinformatics methods to identify the GPX gene in peppers at the whole-genome level. In addition, we analyzed the physical and chemical properties, protein secondary/tertiary structures, chromosome localization, collinearity, phylogenetic relationship, gene structure, conserved motifs, cis elements, protein interaction network, GO enrichment, and subcellular localization of the identified GPX family members. Furthermore, we conducted expression profiling and enzyme activity change analysis to understand their response to abiotic stress (i.e., cold, drought, and salt) and ABA signaling.

## 2. Results

### 2.1. Identification and Basic Information about the CaGPX Gene Family

Using eight *AtGPX* protein sequences from *Arabidopsis* as a reference for BLASTP alignment, a total of eight putative GPX genes were identified in the entire genome of the pepper “CM334” and named *CaGPX1*-*8* according to their position on the chromosome (Table 1). The peptide length encoded by *CaGPXs* is 169 (*CaGPX1*)-250 (*CaGPX2*) amino acids (Table 1). The theoretical molecular weight of the protein ranges from 18.81 to 26.54 kDa, and the calculated isoelectric point (PI) values range from 4.98 to 9.56. Only *CaGPX6* (PI = 5.05) and *CaGPX7* (PI = 4.98) are acidic proteins, while the others are alkaline proteins (Table 1). In addition, all protein instability indices encoded by *CaGPX* genes were less than 40, and the aliphatic index was greater than 90, indicating good protein stability (Table 1). Moreover, the hydropathicity of *CaGPX* proteins is less than one, indicating that they are all hydrophilic proteins (Table 1).

### 2.2. Prediction of Subcellular Localization and Protein Secondary and Tertiary Structure of the CaGPX Gene Family

The results of subcellular localization prediction (Appendix A) showed that *CaGPX* proteins were located in the chloroplasts (*CaGPX1*, *4*, *5*, *7*, *8*), nucleus (*CaGPX2*), and cytoplasm (*CaGPX3*, *6*), respectively. Secondary structure prediction (Appendix A) showed that the CaGPX proteins were mainly composed of α-helices and random coils, which accounted for a large proportion of the protein structure with a small difference in proportion. This was followed by the extended strand structure, which accounted for between 22.35% and 29.6%, while the beta-turn structure accounted for no more than 12%. From the perspective of tertiary structure (Appendix A), CaGPX1 and CaGPX5-8 had a similar structure to AtGPX2, 3, 5, 7, and 8, while CaGPX2-4 had a similar structure to AtGPX1, 4, and 6. In general, the tertiary structure of GPX protein family members in pepper was similar and conserved, basically consistent with the predicted results of the secondary structure. However, the spatial polymerization angle of some proteins was different.

### 2.3. Predicted Properties of CaGPX Proteins

To analyze the characteristics of all eight CaGPX proteins, we compared the GPX amino acid sequences of peppers, *Arabidopsis*, and rice (Figure 1). The results showed that the GPXs of all three species contained three completely conserved Cys residues (marked with a red triangle in Figure 1) and three conserved structural domains that exist in most plants and mammals [34]: GPX domain I (GKVLLIVNVASXCG), GPX domain II (ILAFPCNQF), and domain III (WNFXKFL) (marked with black boxes in Figure 2). Furthermore, there were other conserved sequences in the three species, such as CT(R/I)FKE(Y/F)P(I/V)F, K(V/I)(D/E/R)(V/L)NG, P(L/I)Y(K/E/Q/N/V)FLK, and V(V/I//S)(E/D/Q)RY(P/S/A/G)(P/T)TSP (marked with red boxes in Figure 2). In addition to the known catalytic site Cys residues, three high-potential catalytic sites—Gln, Trp, and Asn—were also found (marked with a blue triangle in Figure 1).

### 2.4. Chromosome Distribution and Collinearity Analysis of the CaGPX Gene Family

TBtools (V 1.118) and MCScanX software (V 2.096) were used to visualize the chromosomal location information of *CaGPX* genes. As shown in Figure 2, the eight *CaGPX* genes were only localized on four chromosomes of pepper, including two on chromosome 1 (*CaGPX1*, *2*), two on chromosome 6 (*CaGPX3*, *4*), one on chromosome 9 (*CaGPX5*), and two on chromosome 12 (*CaGPX4*, *5*). Notably, *CaGPX8* is not anchored to the chromosome but is in the scaffold. Additionally, *CaGPX6* and *CaGPX7* belong to a pair of tandem repeat genes (represented by blue lines in the figure). However, no fragment replication gene pairs were found in the eight *GPX* genes of pepper. Collinearity analysis (Figure 3) found that the *GPX* gene between pepper and *Arabidopsis* had high homology. There were four pairs of collinear gene pairs between three *CaGPX* genes and four *AtGPX* genes, among which *CaGPX1* and *AtGPX6*/*8* had collinear relationships, respectively; *CaGPX3* and *AtGPX3* had a collinear relationship; and *CaGPX5* and *AtGPX5* had collinear relationships.

The nonsynonymous substitution to synonymous substitution (Ka/Ks) ratio is an important indicator for evaluating gene replication events and selection pressure [35]. Therefore, we used KaKs_Calculator 2.0 software to calculate the Ka/Ks values of these three pairs of collinear genes (Appendix A) and found that the ratio of Ka/Ks was less than 1.0, indicating that they mainly evolved under the influence of purification selection, with repeated events occurring 230.2 (*CaGPX1*/*AtGPX8*), 196.0 (*CaGPX1*/*AtGPX6*), 131.69 (*CaGPX3*/*AtGPX3*), and 159.8 million years ago (*CaGPX5*/*AtGPX5*) [36].

### 2.5. Phylogenetic Relationships of the GPX Gene Family

To study the evolutionary relationship between *GPX* genes from different species and *CaGPX* genes in peppers, we constructed an unrooted phylogenetic tree using *GPX* protein sequences from seven species, including pepper, *Arabidopsis*, cucumber, watermelon, *Rhodiola crenulata*, apple, and rice. As a result, a total of 43 GPX proteins from 7 species were clustered into four groups according to their sequence similarity and named Group I–Group IV (Figure 4). Group I consisted of 9 members, Group II consisted of 15 members, Group III consisted of 12 members, and Group IV consisted of 7 members. Among them, CaGPX8 fell into Group I; CaGPX2, 3, 6, and 7 fell into Group II; CaGPX4 and 5 fell into Group III, and CaGPX1 fell into Group IV. Overall, the GPX protein sequences clustered into the same group had a high similarity. In addition, it was worth noting that each group contains GPX proteins from different species, and the pepper GPX proteins have a strong phylogenetic relationship with *Arabidopsis* and rice, indicating that each GPX has homology in several other plant species, so they may have similar functions in different plants.

### 2.6. Analysis of CaGPXs Conserved Motifs and Gene Structures

Given that the pattern diversity of exon/intron structure and protein domains plays a crucial role in the evolution of gene families, we conducted an analysis of the phylogenetic relationships (Figure 5a), conservative motifs (Figure 5b; Appendix A), and gene structures (Figure 5c) of GPX proteins in pepper, *Arabidopsis*, and rice. The results showed that all GPX protein sequences contained three conserved motifs (named motifs 1, 2, and 3). These conserved motifs were similar in GPX proteins of the same group. Additionally, another special motif (named motifs 4-10) was detected in some groups (Figure 5; Appendix A). For example, *OsGPX4* and *CaGPX2* in Group I had motif 7; *AtGPX1*, *7* and *CaGPX8* had motif 5 but were not found in other groups; and motif 10 was limited to two genes (*OsGPX1* and *OsGPX3*) in Group IV. It is worth noting that all members in Group II only had motifs 1–3 (except for *CaGPX3* and *AtGPX3*, which had motif 6); and only motifs 1–4 existed among all members of Group III. In addition, motif 9 (only present in *AtGPX7* and *CaGPX2*) and motif 8 were specific to Group I. From the perspective of gene structure (Figure 5c), the *GPX* gene structure of each group was generally similar, but there were differences in the exon/intron arrangement of some genes. For example, all *GPX* genes have 5–6 exons. In Group I, all members had six exons, and the sequence lengths were generally similar. It is worth noting that the *CaGPX7* sequence in Group II had the longest length (exceeding 15 kb); The member sequences in Group IV were similar in length, *OsGPX1* had only five exons, and other members had similar structures. Overall, *GPX* genes in the same subgroup had similar structures and conserved motif distributions, indicating that *GPX* proteins contain highly conserved amino acid residues and that *GPX* members in the same cluster may have similar roles.

### 2.7. Cis Element Analysis of the CaGPX Promoter in Pepper

Since *GPX* genes play important roles in response to various stresses, we utilized PlantCARE to analyze the cis elements in the pre-2000 bp region of the *CaGPX* gene promoter to explore their key roles in stress and hormone signal transduction. From the promoter region of the *GPX* genes of the target species, we screened 25 putative cis-acting elements and divided them into four categories, including abiotic/biotic stress elements, hormone response elements, light response elements, and growth/development-related elements (Figure 6; Appendix A). The results (Figure 6 and Figure 7) showed that among abiotic/biotic stress elements, hypoxia (ARE) elements were present in all *CaGPX* genes, and in drought-related elements (MYC, as-1, MBS), MYC elements were the most abundant, which was not found in *CaGPX6*. In addition, other cis-acting elements were specifically present in *CaGPX* genes, such as defense and stress responsiveness elements (TC-rich repeats), which only existed in *CaGPX1*, *3*, and *4*; hypoxia (GC-motif) elements only exist in *CaGPX4*, *5*, and *7*. Notably, among the hormone-responsive elements, elements related to abscisic acid responsiveness (ABRE and AAGAA-motif) were the most diverse and present in all *CaGPX* genes. The next was the ethylene-responsive element (ERE), which only exists in *CaGPX5*, *6*, and *8*. However, the gibberellin-responsive element was only present in *CaGPX3* and was not found in other *CaGPX* genes. Finally, light response elements were relatively abundant in all *CaGPX* genes, with the GATA motif only existing in *CaGPX2* and *8*, while growth and development elements were specifically present in *CaGPX8* genes.

### 2.8. Functional Annotation Evaluation of CaGPX Genes

To further understand and distinguish the role of *CaGPX* genes, we utilized the eggNOG online website to perform GO annotation and enrichment analysis of their functions. The biological process (BP) enrichment results (Appendix A) showed that the *CaGPX* gene was mainly involved in cellular response to chemical stimulus (GO:0070887), detoxification (GO:0098754), cellular oxidant detoxification (GO:0098869), response to stimulus (GO:0050896), response to stress (GO:0006950), and cellular response to stimulus (GO:0051716), etc. The cellular component (CC) enrichment showed that *CaGPX* genes were mainly associated with cytosol (GO:0005829) and cytoplasm (GO:0005737). It is worth noting that molecular function (MF) was enriched in six GO terms, including antioxidant activity (GO:0016209), peroxidase activity (GO:0004601), glutathione peroxidase activity (GO:0004602), oxidoreductase activity (GO:0016491), and oxidoreductase activity, acting on peroxide as acceptor (GO:0016684), and catalytic activity (GO:0003824). These terms indicate that the *CaGPX* gene plays a role in reactive oxygen scavenging and antioxidant defense.

### 2.9. Analysis of the Protein Interaction Network of CaGPX Family Genes

To further understand the function of *CaGPX* genes in pepper, we utilized the STRING website to construct a potential protein–protein interaction network (PPI) between CaGPX proteins and other coding proteins in peppers. The results (Figure 8; Appendix A) showed that the PPI network involves a total of 28 nodes and 392 edges. CaGPX1, T459_13041, GR1, T459_08909, T459_08876, and DHAR were the most significant genes with a betweenness centrality value greater than 20. In the PPI network, there were a total of 19 encoded proteins interacting with CaGPX proteins, among which three were glutathione reductase (GR1, T459_08876, and T459_08909); DHAR and T459_29115 belong to the GST superfamily, and T459_12719, T459_16601, T459_28703 and T459_32139 belong to the peroxidase family; two types were monodehydroascorbate reductase (T459_00229, T459_06854), two types were gamma-glutamyl transpeptidase 2, 3 (T459_15015, T459_24309), and two types were 2-Cys peroxiredoxin BAS1(T459_03397, T459_26258). The other three types were MRG domain-containing protein (T459_13041), glutathione synthetase (T459_22335), and thioredoxin reductase (T459_26495).

### 2.10. Tissue-Specific Expression Profiles of the CaGPX Gene in Peppers

To explore whether the *GPX* gene family plays a role in the tissue development of pepper, the expression profiles of *CaGPXs* in the roots, stems, leaves, pericarp, and placenta were observed using RNA-seq data from the pepper cultivar “CM334”. The results (Figure 9) showed that the expression of *GPX* genes was specific in different tissues of peppers, and *CaGPX1*, *4*, and *6* were highly expressed in all tissues. Similarly, the expression of *CaGPX8* was higher in leaves, stems, pericarp (6 d, 16 d, and 25 d after anthesis, green ripening, color change stage, and 5 d, 10 d after color change), and placentas (16 d after anthesis and 5 d and 10 d after color change), while it was less expressed in other tissues. By contrast, compared with other genes, *CaGPX2* and *CaGPX3* were expressed at lower levels in all tissues. Therefore, we speculated that *CaGPX1*, *CaGPX4,* and *CaGPX6* play a greater role than other genes in the development processes of pepper.

### 2.11. Changes in the Relative Expression of the CaGPX Gene under Osmotic Stress and ABA Treatment

To further understand the expression of the *CaGPX* gene under different abiotic stress conditions, qRT-PCR was used to analyze the expression of the eight *CaGPX* genes under ABA, cold, drought, and salt stress (Figure 10). Under 100 μmol/L ABA treatment (Figure 10a), the expression of most *CaGPX* genes reached the highest level at 12 h after treatment, with *CaGPX1* being the most significant, 53-fold higher than that at 0 h (CK), followed by *CaGPX6*, which was 29-fold higher than that at 0 h. However, the expression of *CaGPX4* reached its maximum at 6 h, which was 81.5-fold higher than that at CK. By contrast, after 24 h of treatment, the expression levels of *CaGPX5*, *7*, and *8* were downregulated, and the expression level of *CaGPX7* was the most significantly downregulated (0.25). Notably, *CaGPX3* expression was downregulated at all other times and increased only 12 h after treatment, rising 1.7-fold over CK. Under cold stress (Figure 10b), the expression of *CaGPX4* and *CaGPX6* were significantly upregulated, reaching a maximum at 6 h, which was 65-fold and 16-fold higher than that at 0 h, respectively; *CaGPX1* reached the maximum value at 12 h, which was more than 19 times higher than that at 0 h; *CaGPX2* and *CaGPX3* reached the maximum value at 24 h, which was 9 times and 2.5 times higher than at 0 h, respectively; and *CaGPX8* peaked at 12 h after treatment, 9 times that at 0 h. However, although the upregulation of *CaGPX7* expression level reached a significant level after treatment, the highest was only 1.7 times that of 0 h. Notably, the expression levels of all *CaGPX* genes increased first and then decreased under drought stress (Figure 10c) and reached a maximum at 6 h (except for *CaGPX5* and *CaGPX3*), which were 92-, 5-, 54-, 21-, 9-, and 23-fold higher than those at 0 h, respectively. The expression levels of *CaGPX5* and *CaGPX8* decreased to 0.7 at 24 h. Similarly, the expression trends of *CaGPX1* and *CaGPX4* under salt stress were the same as those under drought stress (Figure 10d), reaching their maxima at 6 h, which was 55 times and 74 times higher than that at 0 h, respectively. The expression levels of *CaGPX5* and *CaGPX6* were the highest at 9 h, which was 9.3 times and 9.6 times higher than that at 0 h, respectively. The above results indicate that *CaGPX1*, *4* and *CaGPX6* play an important role in alleviating abiotic stress in pepper.

### 2.12. Subcellular Localization Analysis

Subcellular localization of genes helps to understand the working position of proteins. Because the relative expression levels of *CaGPX1* and *CaGPX4* were significantly upregulated under different stress treatments, and the expression trends were also similar. Therefore, in order to verify the results of subcellular prediction, we analyzed the instantaneous expression of CaGPX1 and CaGPX4 proteins in tobacco leaves using Agrobacterium-mediated methods and observed the fluorescence signals by laser confocal microscopy. The results showed that the strong fluorescence signal was localized in chloroplasts (Figure 11), which was also consistent with our prediction.

### 2.13. Changes in Glutathione Peroxidase Activity (GSH-Px) and Morphological Characteristics of Pepper Seedlings under Osmotic Stress and ABA Treatment

The activity of glutathione peroxidase and the phenotypic changes in pepper seedlings under different treatments are shown in Figure 12 and Appendix A, respectively. The results showed that the activity of glutathione antioxidant enzymes showed a trend of first increasing and then decreasing under different treatments. The enzyme activity under ABA (Figure 12a) and cold treatment (Figure 12b) peaked at 12 h, 3.2 and 2.2 times that at 0 h, respectively, followed by 9 h, 2.9 and 1.7 times that at 0 h, respectively. The maximum enzyme activity under PEG (Figure 12c) and NaCl treatment (Figure 12d) was reached at 6 h, which was 2.9 and 2.3 times that of 0 h, respectively; 9 h arrived next, with 2.5 and 1.8 times the values at 0 h, respectively. Additionally, the variations in gene expression are essentially consistent with the variations in enzyme activity, and the degree of wilting of pepper plants also increased with the lengthening of the treatment period, but the phenotype of plants receiving ABA treatment was good (Appendix A).

## 3. Discussion

As a consequence of the various biotic or abiotic stress processes that plants can face, a large amount of ROS accumulates in plant cells, leading to the damage of biological macromolecules and cell death. In this sense, the protein encoded by the *GPX* gene plays an important role in protecting plant cells from oxidative damage caused by ROS accumulation [2,37,38]. Therefore, a systematic and comprehensive analysis of the pepper *GPX* gene family and understanding their response to a variety of stress conditions is a justified scientific question to address [17]. In our study, a total of eight *CaGPX* genes were identified from the “CM334” pepper genome. All eight *CaGPX* genes encode hydrophilic proteins with stable structures. *CaGPX6* and *CaGPX7* encode proteins with isoelectric points lower than the others (Table 1). This result was similar to that in rapeseed and watermelon. In *BnGPX* and *ClGPX*, all the proteins encoded by genes are hydrophilic proteins with stable structures, and they also contain two acidic proteins, while the remaining proteins are alkaline [12,39]. Subcellular localization prediction can locate a certain protein or expression product at a specific location in the cell, providing a research direction for understanding the mechanism of gene action [40]. Like *CsGPXs* in cucumber and *GhGPXs* in cotton, *CaGPXs* were mainly located in the cytoplasm, nucleus, and chloroplast (Appendix A), indicating that *GPX* genes act on these organelles and protect them from damage during ROS production. In addition, H_2_O_2_ produced in chloroplast PSI is one of the main sources of intracellular ROS, so *CaGPXs* located in chloroplasts may also be involved in regulating the generation of H_2_O_2_ in chloroplasts [41].

The amino acid sequence analysis of CaGPX proteins, AtGPX proteins, and OsGPX proteins showed that their encoded proteins have three typical GPX domains and three conserved Cys residues of GPX active centers (Figure 1), and Cys residues are specific structures of selenium-independent plant GPXs [42]. The first two Cys residues were located within domain I and domain II, respectively, and the third was located outside the domain. These results indicate that biological GPX proteins are highly similar in sequence, and the key active site Cys residues are crucial to the perception of redox potential [11,43]. Chromosome mapping showed that the eight *CaGPX* genes were distributed on four chromosomes of pepper, and *CaGPX6*/*CaGPX7* were a pair of tandem repeat gene pairs (Figure 2). Segment repetition, tandem repetition, and translocation events are the primary sources of gene family amplification [44], which is a fundamental evolutionary process in the genome that assists plants in adapting to varied environmental conditions [45]. The collinearity analysis of the *CaGPX* gene and *AtGPX* gene showed that there were four pairs of collinearity relationships between two *CaGPX* genes and three *AtGPX* genes (Figure 3), and they belonged to purification selection during evolution (Appendix A), indicating that the *GPX* gene had a common ancestor and was highly conserved in the evolutionary process. These results also indicate that tandem duplication events play an important role in the *GPX* gene replication process in peppers.

The phylogenetic tree shows that the *GPX* genes from peppers and other species were divided into four groups, which was consistent with the classification in cucumber [46], cotton [26], *Thellungiella salsuginea* [47], rapeseed [12], and bread wheat [17]. By observing the phylogenetic tree, similarities and differences between different species can be analyzed. Our results show that each group contains GPX proteins from different species, and pepper GPX proteins are closely branched with *Arabidopsis* and rice on the phylogenetic tree (Figure 4). This suggests that these gene families have undergone numerous evolutionary processes, including gene replication, division, and modification, leading to the emergence of numerous isoforms and potential orthologs in various species. Simultaneously, in Figure 5a, there were also four clusters discernible in the grouping classifications of *CaGPXs*, *AtGPXs*, and *OsGPXs*. *GPX* genes aggregated into the same group have similar conserved motifs and gene structures (Figure 5). Three conserved motifs (1–3) are present in all *GPX* genes in pepper, *Arabidopsis*, and rice, while some other conserved motifs are limited to particular groups or species of GPX proteins (Figure 5b), suggesting that these motifs may support the maintenance of particular *GPX* functions. The structure of gene exons and introns is an important marker to reveal the evolutionary relationship between gene family members [48]. Gene structure analysis showed that *CaGPXs*, *AtGPXs*, and *OsGPXs* almost all had five introns and six exons (Figure 5c), which was similar to the results in cucumber [46], *Thellungiella salsuginea* [47], and cotton [26]. Their *GPX* genes also have five introns and six exons. The above results indicate that *GPX* genes were highly conserved during evolution and may have similar functions.

Cis-acting elements play an important role in the regulation of gene expression, which can bind to transcription factors and regulate gene transcription [49,50]. By analyzing the cis elements in the promoter region of the *CaGPXs* gene, we can better understand its regulatory mechanism and infer the function of the *CaGPXs* gene and its influence on the development of pepper. The results showed that the identified cis elements were mainly divided into four categories, i.e., stress response elements, plant hormone response elements, growth and development elements, and light response elements (Figure 7; Appendix A). The types and quantities of elements contained in different *CaGPXs* were different, indicating that the *CaGPX* genes have different levels of stress and hormone responses, so it was speculated that there were functional differences among *CaGPXs*. Additionally, GO enrichment annotations demonstrated that *CaGPX* genes are involved in many stress responses and possess peroxidase activity (Appendix A), and previous reports have also suggested that *GPX* genes are involved in several different stress responses [16,27,47]. These results indicate the importance of *CaGPX* genes in the growth and development of peppers.

The results of the interaction network show that GPX proteins all interact with each other, and there was an interaction relationship with the GST superfamily, peroxidase family, glutathione reductase, and glutathione synthetase (Figure 8; Appendix A), which was not addressed in Wang et al.’s study [51] or in previous reports [26,27,39,46]. Since the interaction networks of multiple GPX proteins were highly crossed and clustered into a very large network, we speculated that GPX proteins may be functionally related and may interact with the other families to participate in the clearance of reactive oxygen species in peppers.

Understanding the expression patterns of genes in different plant tissues is important for studying gene function and biological development [52]. The tissue-specific expression profile of *CaGPX* genes showed that during the growth and development of peppers, the expression of some genes had obvious tissue specificity (Figure 9). In cotton, *GhGPX2* was highly expressed in roots and stems, *GhGPX13* was highly expressed in flowers but lower in other tissues, and *GhGPX1*, *3*, and *5* and *GhGPX8* were also highly expressed in flowers [26]. In watermelon, all *ClGPX* genes were highly expressed in flowers and fruits, and *ClGPX3* was also highly expressed in mature leaves [39]. In cucumber, *CsGPX1*, *CsGPX3*, *CsGPX5*, and *CsGPX6* were highly expressed in the tendrils and tendril bases but were expressed at low levels in other tissues [46]. These results indicate that the expression patterns of *GPX* genes vary in different plant tissues, and they play their respective potential roles in plant development.

Similar to the tissue-specific expression results, *CaGPX1*, *4* and *CaGPX6* expression levels were significantly upregulated under ABA and abiotic stress (Figure 10). However, the expression levels of other genes only changed during certain periods after treatment. These results indicate that *GPX* genes exhibit complex expression patterns in stress treatments and thus perform different functions. At the same time, studies have shown that some members of the GPX gene family can be regulated through ABA-dependent signaling pathways [31], in which *AtGPX3* can act as an H_2_O_2_ scavenger in ABA and drought stress signals to transmit H_2_O_2_ signals, thereby removing H_2_O_2_ [53]. In watermelon and cucumber, almost all *GPX* genes were significantly upregulated under ABA, cold, salt, and drought treatments [39,46]. In wheat, the *GPX* genes *W69* and *W106* play an important role in response to salt, drought, hydrogen peroxide, and ABA, and their overexpression enhances salt tolerance in transgenic *Arabidopsis* [31]. In *T. salsuginea*, almost all *TsGPX* genes were significantly upregulated at least at one time point under salt and drought stress [47]. Furthermore, the variations in glutathione peroxidase (GSH-Px) activity in peppers are essentially compatible with the variations in gene expression levels, and the variations in glutathione peroxidase activity in peppers were also described for the first time in this study. The above results indicate that the CaGPX gene may enhance the stress resistance of pepper by regulating ABA signaling pathways, and CaGPX1, 4 and CaGPX6 may be more important in this process. At the same time, our findings also lay the foundation for further research on the function of the CaGPX gene family in abiotic stress and ABA responses.

## 4. Materials and Methods

### 4.1. Genome-Wide Identification of CaGPX Genes in Pepper

The protein sequences of 8 *GPX* genes in *Arabidopsis* were downloaded from the *Arabidopsis* information resource website (https://www.arabidopsis.org/, accessed on 6 November 2023) [54]. The “CM334” (*C*. *annuum*) genomic data were downloaded from the pepper website (http://peppergenome.snu.ac.kr/download.php, accessed on 6 November 2023) [32]. Then, BLASTp comparisons were performed using the amino acid sequences of eight *Arabidopsis GPXs* as a query, with the E-value < E^−10^. Next, we used the hidden Markov model (HMM) profile of the *GPX* structural domain in the Pfam database (https://www.ebi.ac.uk/interpro/, accessed on 6 November 2023) [55] to identify candidate *GPX* sequences (Pfam: PF00255). Finally, the NCBI-CDD databases (https://www.ncbi.nlm.nih.gov/cdd/, accessed on 6 November 2023) [56] were used for domain identification of candidate gene sequence structures. Candidate genes that did not contain the specific domain of the *GPX* gene (Pfam: PF00255) were manually removed.

### 4.2. Sequence Information and Structural Analysis of the Pepper GPX Gene Family

The subcellular localization of GPX proteins in pepper was predicted using the WoLf PSORT online website (https://wolfpsort.hgc.jp/, accessed on 6 November 2023) [57]. ExPASy (http://expasy.org/, accessed on 6 November 2023) was used to identify the number of amino acids, the isoelectric point, and other physical and chemical properties of CaGPX proteins [58]. Then, the secondary structure of the pepper GPX protein was predicted by the online site SOPMA (https://npsa-prabi.ibcp.fr/cgi-bin/npsa_automat.pl?page=npsa%20_sopma.html, accessed on 6 November 2023) [59]. Finally, the tertiary structures of pepper GPX and *Arabidopsis* GPX proteins were predicted by Phyre2 (http://www.sbg.bio.ic.ac.uk/, accessed on 6 November 2023) [60] and visualized using PyMOL software (V 2.60a0) [61].

### 4.3. Analysis of Chromosome Location and Collinearity

TBtools software (V 1.118) was used to map and visualize the position of the *GPX* genes identified in the 12 chromosomes of pepper [62]. Homology was searched using MCScanX, protein-coding genes in the pepper genome were compared with the protein-coding genes in the *Arabidopsis* genome using BLASTp, and the search threshold was set to an E-value < E^−5^. Other parameters default. Whole-genome BLASTp results were used to compute collinear blocks for all possible pairs of chromosomes and scaffolds [63]. Finally, the *GPX* gene colinear pairs of pepper and *Arabidopsis* were highlighted using TBtools (V 1.118) [62]. In addition, we used KaKs_Calculator 2.0 software to calculate the Ka/Ks ratio of CaGPX genes that have a collinear relationship with *Arabidopsis* [64].

### 4.4. Multiple Sequence Alignments and Construction of Phylogenetic Tree

To understand the evolutionary relationship of the *CsGPX* gene family, we used the muscle sequence alignment method in MEGA 11 software (V 11.08) to compare the GPX protein sequences of seven species (Appendix A), including pepper, *Arabidopsis*, cucumber, watermelon, *Rhodiola crenulata*, apple, and rice. Then, we selected the maximum likelihood (ML) method to construct a phylogenetic tree. The amino acid substitution model was LG+G (Additional Appendix A), and the bootstrap values were calculated for 1000 replicates [65]. Next, the constructed tree was beautified using the EvolView online website (https://www.evolgenius.info/evolview/#/treeview, accessed on 6 November 2023) [66]. Finally, a sequence alignment representation of the GPX protein sequences of pepper, *Arabidopsis*, and rice was generated using Jalview software (V 1.8.3) [67].

### 4.5. Analysis of Gene Exon–Intron Structures and Protein Conserved Motifs

The structure of *GPX* genes in pepper, *Arabidopsis*, and rice was analyzed by GSDS 2.0 (http://gsds.gao-lab.org/, accessed on 6 November 2023) [68]. Then, the conserved motifs of the *GPX* genes in pepper, *Arabidopsis,* and rice were analyzed by MEME version 5.5.1 (http://meme-suite.org/tools/meme, accessed on 6 November 2023) [69]. The maximum number of motifs was set as 10, and the remaining parameters were set as default values.

### 4.6. Analysis of Cis-Acting Elements in CaGPX Gene Promoters

To understand the inferred cis-elements in the *CaGPX* promoter, the first 2000 bp upstream sequence of 8 identified *CaGPX* gene initiation codons (ATG) was extracted using TBtools (V 1.118) [62]. Then, each extracted gene promoter sequence was submitted to the PlantCARE website (http://bioinformatics.psb.ugent.be/webtools/plantcare/html/, accessed on 6 November 2023) [70] for cis-acting element prediction, and a figure of its distribution into the promoters was generated using TBtools (V 1.118) [62].

### 4.7. Tissue Expression Analysis of GPX Genes

To investigate the specific expression of the *CaGPX* gene in different tissues of peppers, this study utilized transcriptome data from “CM334” and normalized the fragment count of each gene using the million fragments per thousand bases method (FPKM) [32]. Finally, data conversion was performed using the log2 (FPKM + 1) method, and TBtools software (V1.118) was used to obtain the expression profiles of the *GPX* genes in various tissues [62].

### 4.8. Functional Annotation Study of CaGPX Genes and Protein Interaction Network Prediction

CaGPX protein sequences were uploaded to the eggNOG website (http://eggnog-mapper.embl.de/, accessed on 6 November 2023) to perform their gene ontology (GO) annotation [71]. Additionally, TBtools (V 1.118) was used for GO enrichment analysis and visualization. The STRING website (https://cn.string-db.org/cgi/, accessed on 6 November 2023) [72] was used to predict protein–protein interaction networks using the five CsGPX protein sequences as targets.

### 4.9. Test Materials and Treatment

This study used the Longjiao 10 pepper variety (*Capsicum chinense*). Pepper seedlings were cultured in an artificial climate incubator under photoperiod conditions of 26/20 °C day/night and 16/8 h day/night with a light intensity of 300 μmol m^−2^ s^−1^ for 6–8 true leaf stages, and various treatments were performed. Plants of uniform growth were then treated with four treatments, T1: 100 μmol/L ABA; T2: cold stress (4 °C); T3: 20% PEG; and T4: 200 mmol/L NaCl. All treatments were performed simultaneously, and leaves were harvested from the same location of the seedlings after treatment for 0, 3, 6, 9, 12, and 24 h, with three biological replicates. Samples were placed in liquid nitrogen and immediately stored in a refrigerator at −80 °C.

### 4.10. RNA Extraction and Quantitative PCR (qRT-PCR)

Total RNA from pepper leaves was extracted using an RNA extraction kit (Accyrate Biotechnology Co., Ltd., Yiyang, China). With 2 µL RNA as a template, cDNA was obtained by reverse transcription using the Evo M-MLV reverse transcription kit (Accyrate Biotechnology Co., Ltd., China). Using pepper actin as the internal reference gene (NCBI login number: LOC107875540), the coding sequences of *CaGPX* genes were input into the homepage of Shanghai Biology Company (Shanghai, China) for online primer design (Appendix A). qRT-PCR was performed by fluorescence quantitative analysis using the SYBR Green kit (Accyrate Biotechnology Co., Ltd., China). The volume of the reaction system was 20 µL, containing 1 µL cDNA solution, 10 µL 2 SYBR, 2 µL of 10 µM forward and reverse primers, and 7 µL of distilled deionized water. Next, qRT-PCR was performed using the LightCycler^®^ 480 II real-time fluorescence quantitative PCR instrument. The amplification program conditions were as follows: 95 °C for 15 min, 40 cycles of 95 °C for 10 s and 60 °C for 30 s. Each sample was replicated three times.

### 4.11. Subcellular Localization of CaGPX1 and CaGPX4 Proteins

In order to determine the subcellular localization of CaGPX1 and CaGPX4 proteins, we amplified the full length without the stop codon of the *CaGPX1* and *CaGPX4* genes and inserted the amplified fragments into the Pac402 vector. The primer sequences are shown in Appendix A. Subsequently, the fusion vector plasmid was transformed into *A. tumefaciens* (GV3101) using the freeze–thaw method. Then, *A. tumefaciens* was infiltrated into tobacco (*Nicotiana benthamiana*) leaves. After 48 h of dark infection at 25 °C, green fluorescence was observed using a laser-scanning confocal microscope (Olympus FV3000, Tokyo, Japan).

### 4.12. Determination of Glutathione Antioxidant Enzyme Activity

The determination of GSH-Px activity was based on Xue et al.’s method [73], with slight modifications. One enzyme activity unit of glutathione peroxidase is defined as 1 mU of GSH-Px mL^−1^ min^−1^.

### 4.13. Statistical Analysis

The relative expression levels of the *CaGPX* genes were calculated using the 2^−∆∆CT^ method [74], and raw CT values were placed in Additional Appendix A. SPSS 20.0 was used to analyze the relative expression, and Origin 9.0 was used to complete the histogram of relative expression.

## 5. Conclusions

In conclusion, a total of eight members of the *CaGPX* gene family were identified in the “CM334” genome database of pepper. Analysis of their physical and chemical properties, multiple sequence alignment, phylogeny, collinearity, and gene structure revealed that the family had a conserved sequence and stable structure. Furthermore, GO annotation, protein interaction network, subcellular localization, tissue-specific expression profiling, and expression profile analysis under different osmotic stress and ABA treatment suggest that certain *CaGPX* genes play distinct roles in pepper development and stress response and may contribute to the ABA signaling pathway. However, the exact function of these genes remains to be elucidated in future studies. In addition, these findings will lay the foundation for studying the role of the *CaGPX* gene in the development of capsicum, as well as using CRISPR/Cas system overexpression, knockout, and other functional verification schemes.

## Figures and Tables

**Figure 1 ijms-25-08343-f001:**
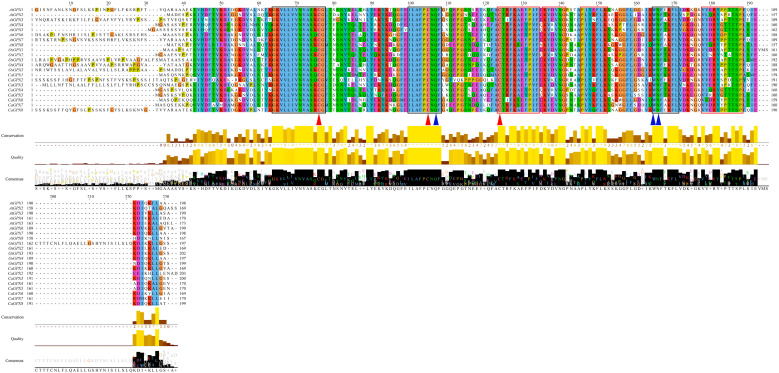
Multiple sequence alignment of GPX protein sequences from pepper, rice, and *Arabidopsis*. Sequence alignment was performed using the ClusterW method in MAGA11 (V 11.08), and the alignment results were visualized with Jalview (V 1.8.3). The bottom two yellow column areas indicate the degree of base similarity, the black part indicates the common base, and the higher the column, the higher the base similarity. The three conserved domains found in most animal and plant GPXs are marked with black boxes; the amino acid residues that form part of the catalytic sites of GPXs are labeled with different colored triangles, including the strictly conserved Cys (C), Gln (Q), Asn (N), and Trp (W). Other conserved domains are marked with a red box.

**Figure 2 ijms-25-08343-f002:**
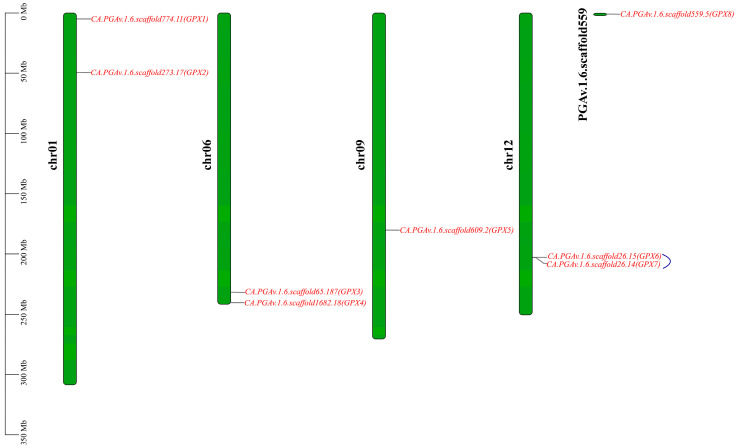
Chromosomal distribution of *CaGPXs*. The green bars represent chromosomes. The chromosome names are shown at the left of each chromosome. The chromosome scale is in millions of bases (Mb) on the left. The blue line represents a tandem repeat gene pair.

**Figure 3 ijms-25-08343-f003:**
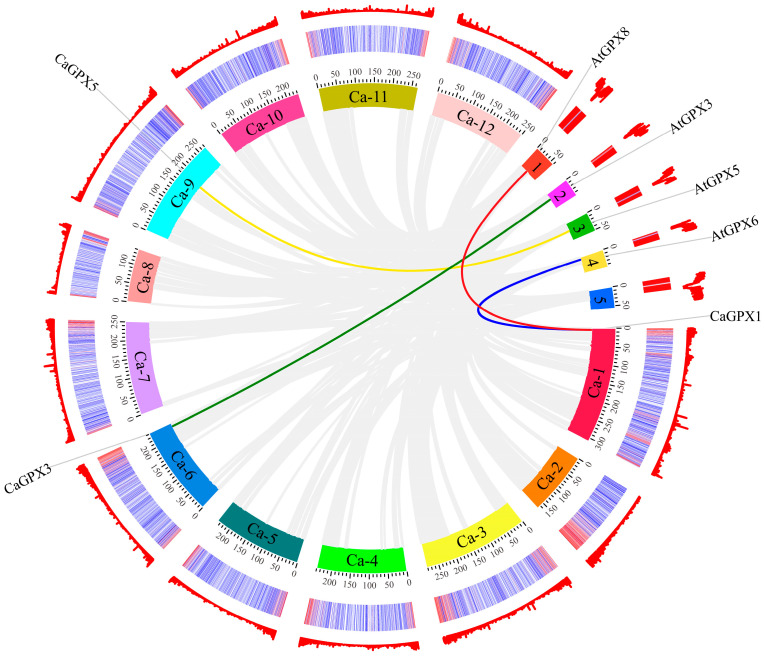
Collinearity analysis of the *GPX* gene family between pepper and *Arabidopsis*. Gray lines denote the collinear blocks between pepper and *Arabidopsis* genomes, and the lines of different colors represent collinear gene pairs of *GPXs* in pepper and *Arabidopsis*. Different colored squares 1–5 represent *Arabidopsis* chromosomes and their length, and different colored Ca-1–Ca-12 represents the pepper chromosome and its length. The outermost two circles represent gene density.

**Figure 4 ijms-25-08343-f004:**
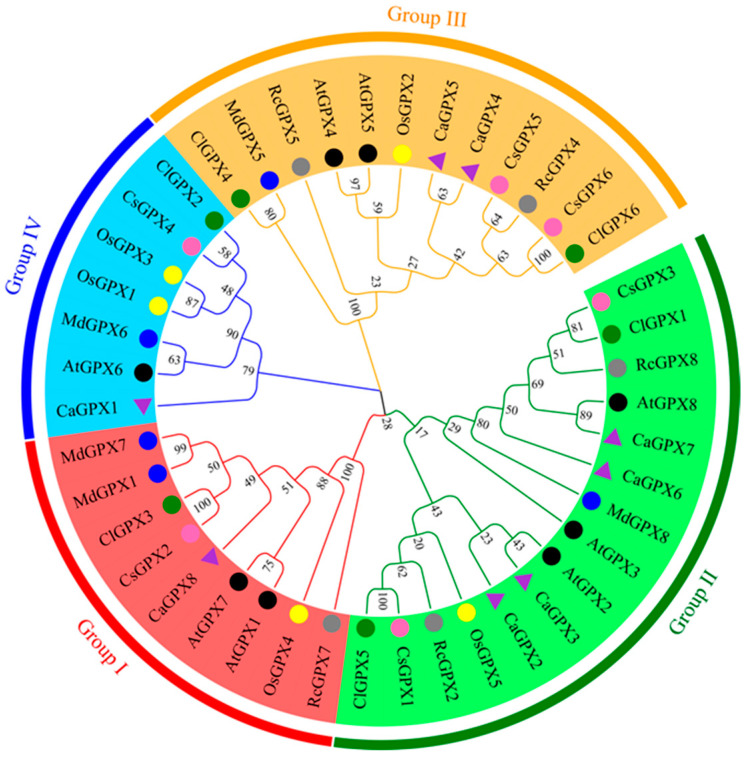
Phylogenic tree of GPX proteins. Purple triangles represent pepper, black circles represent *Arabidopsis*, yellow circles represent rice, pink circles represent cucumber, green circles represent watermelon, gray circles represent *Rhodiola crenulata*, and blue circles represent apple. The phylogenetic tree was constructed by MEGA (V 11.08) using the maximum likelihood method (1000 bootstraps). Different colored areas represent different branches, which are divided into four groups.

**Figure 5 ijms-25-08343-f005:**
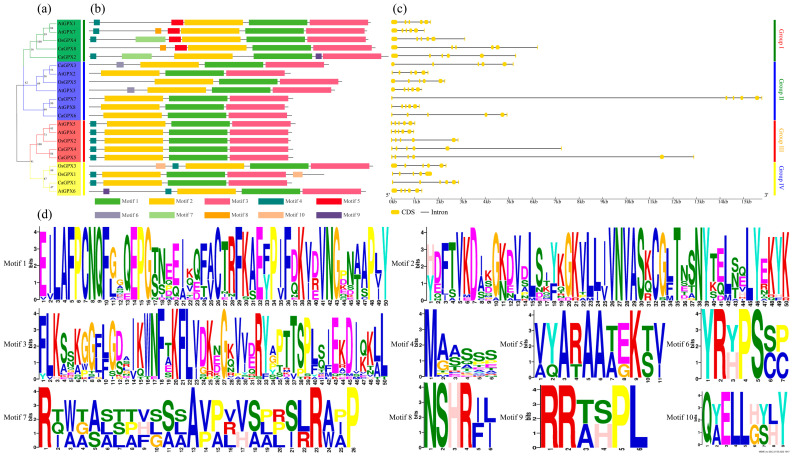
Conserved motif and gene structure analysis of *GPX* family genes. (**a**) The phylogenetic relationship of *GPX* in peppers, *Arabidopsis*, and rice. The tree was constructed by the maximum likelihood method. (**b**) Distributions of conserved motifs in *GPX* genes. Each motif is presented with a particular color. (**c**) The exon/intron structure of the *GPXs*. Yellow represents exon regions and black lines represent introns. (**d**) The pattern identification of ten conserved sequences.

**Figure 6 ijms-25-08343-f006:**
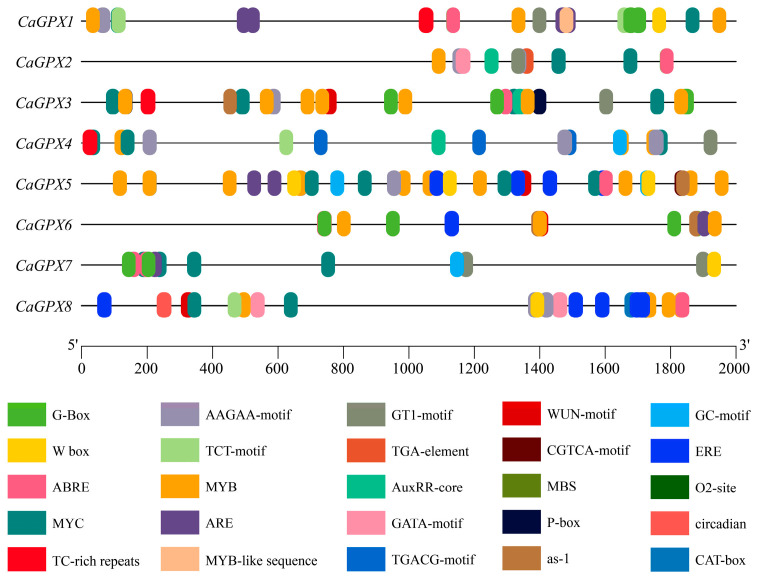
The distribution of cis-regulatory elements predicted in the *CaGPX* gene promoter. Different colored boxes represent different cis elements.

**Figure 7 ijms-25-08343-f007:**
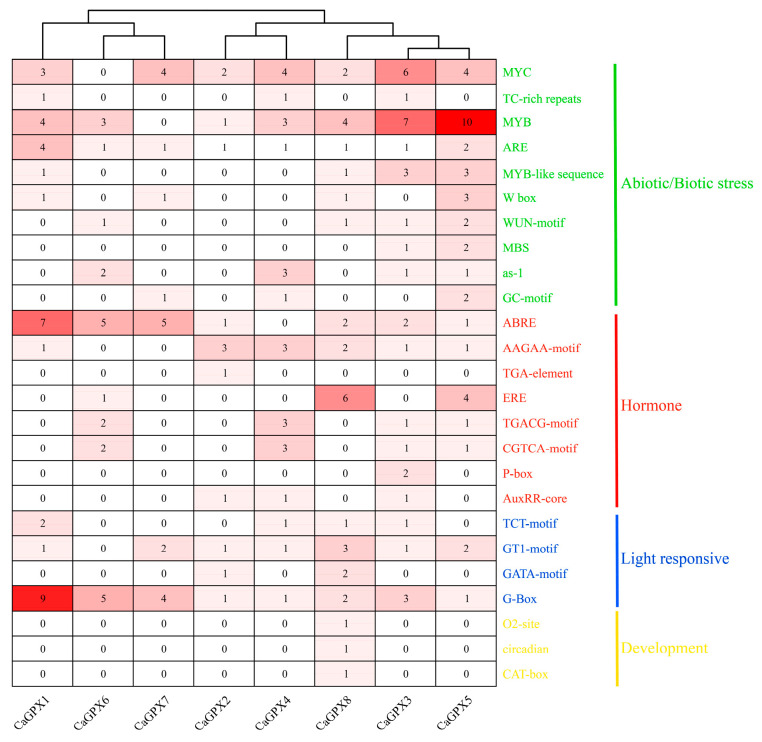
Number statistics and element classification of predicted cis-regulatory elements in the *CaGPX* gene promoter regions. The green part represents the cis-acting elements of biotic/abiotic stress; the red part represents the elements of hormone response; the blue part represents the light response elements, and the yellow part represents the developmental response elements.

**Figure 8 ijms-25-08343-f008:**
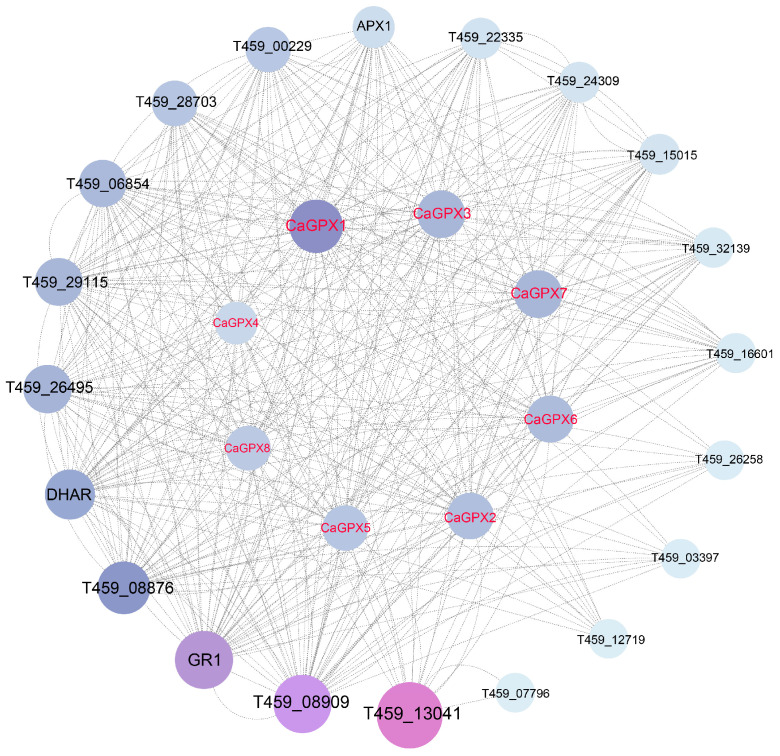
Interaction network analysis of CaGPX proteins. Protein–protein interaction networks were predicted using STRING. The size of the circle and the depth of the color indicate the number of linked nodes and the degree of the protein in the PPI network.

**Figure 9 ijms-25-08343-f009:**
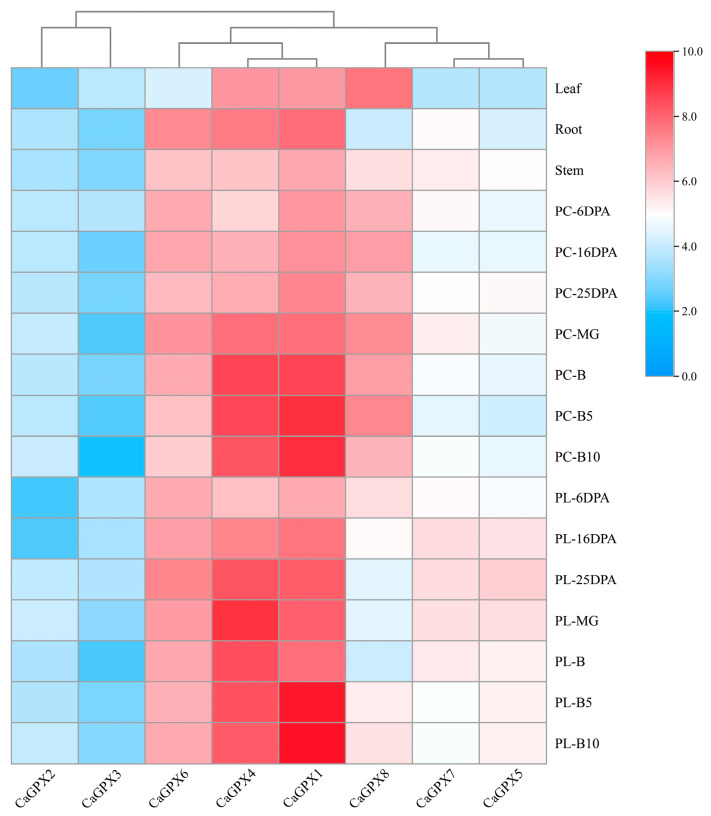
Expression profile analysis of pepper *CaGPX* genes in different tissues and during fruit development. Note: Pericarp (PC) and placenta (PL) at 6, 16, and 25 days postanthesis (DPA); PC and PL at mature green (MG) and breaker (B) stages; PC and PL at 5 and 10 days postbreaker (B5 and B10, respectively). The FPKM values were log2-transformed, and a heatmap was generated using TBtools software (V 1.118). Expression values are shown as a color gradient from low expression (blue) to high expression (red).

**Figure 10 ijms-25-08343-f010:**
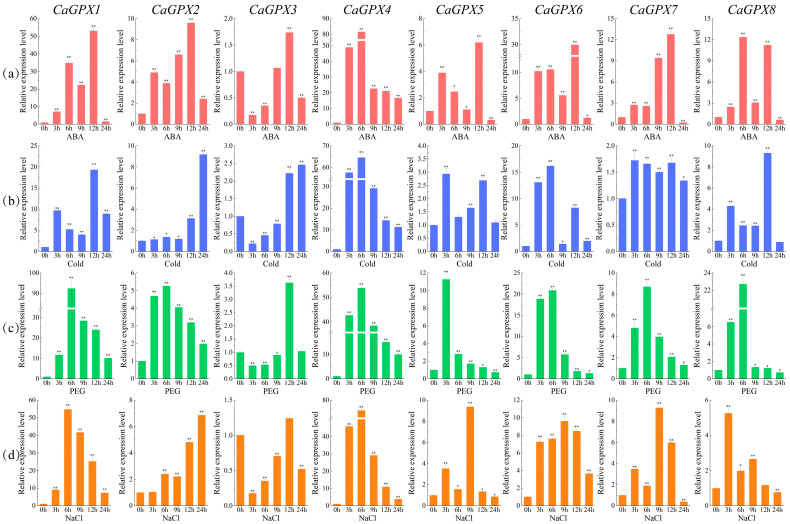
Quantitative real-time PCR analysis of *CaGPX* genes in response to ABA (**a**), cold (**b**), PEG (**c**), and NaCl (**d**) treatments. The relative gene expression was calculated using the 2^−∆∆Ct^ method with CaActin as an internal control, and the value represents the mean ± SE of three biological replicates. Statistical analyses were carried out by a *t*-test to determine the differences in gene expression between 0 h and other treatment times. Asterisks indicate values that are significantly different from CK (0 h) (* *p* < 0.05, ** *p* < 0.01, one-way ANOVA).

**Figure 11 ijms-25-08343-f011:**
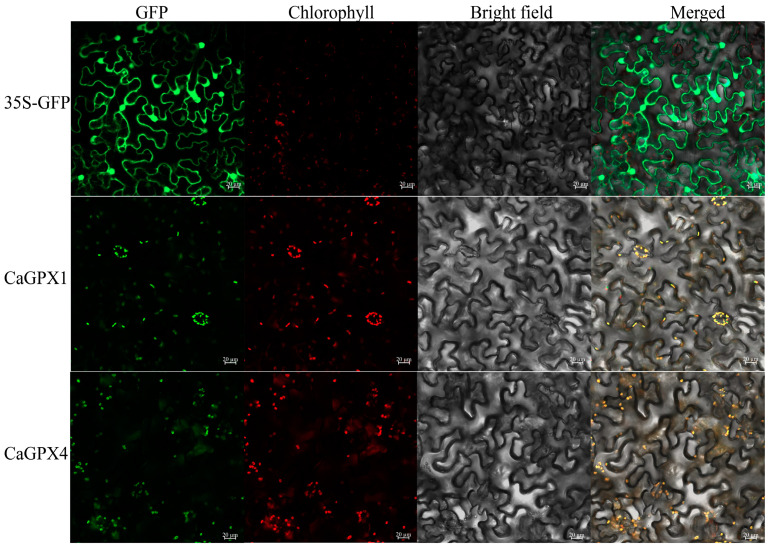
Subcellular localization of CaGPX1 and CaGPX4 proteins in tobacco leaves. GFP signaling displays subcellular localization of Pac402-GFP-CaGPX1 and Pac402-GFP-CaGPX4 proteins. Scale bars represent 20 μm.

**Figure 12 ijms-25-08343-f012:**
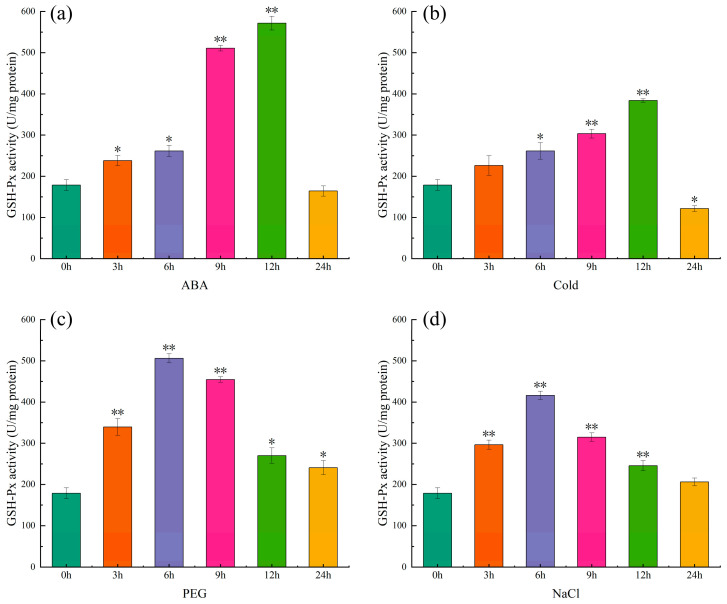
Changes in glutathione peroxidase activity under ABA (**a**), cold (**b**), PEG (**c**), and NaCl (**d**) treatments. The error bar is the standard error of three independent tests, which are statistically analyzed through a *t*-test to determine the difference in enzyme activity between 0 h and other treatment times. Asterisks indicate values that are significantly different from CK (0 h) (* *p* < 0.05, ** *p* < 0.01, one-way ANOVA).

**Table 1 ijms-25-08343-t001:** Genomic information and protein characteristics of 8 *CaGPX* gene family members in pepper.

Gene ID	Gene Name	Chromosome	Gene Location		Amino Acid Number	Molecular Weight (kd)	PI	Instability Index	Aliphatic Index	Grand Average of Hydropathicity
			Start Position	End Position						
CA.PGAv.1.6.scaffold774.11	*GPX1*	1	4864658	4867511	169	18,810.48	8.3	26.88	74.38	−0.402
CA.PGAv.1.6.scaffold273.17	*GPX2*	1	49345165	49350483	250	28,216.25	9.48	37.86	77.2	−0.336
CA.PGAv.1.6.scaffold65.187	*GPX3*	6	231723450	231728659	200	22,682.15	8.19	32.2	86.75	−0.148
CA.PGAv.1.6.scaffold1682.18	*GPX4*	6	241446670	241453933	170	18,929.59	8.87	24.88	76.82	−0.306
CA.PGAv.1.6.scaffold609.2	*GPX5*	9	180100154	180113053	170	18,920.73	9.56	19.42	76.82	−0.301
CA.PGAv.1.6.scaffold26.15	*GPX6*	12	202752908	202757856	169	19,019.56	5.05	28.99	84.2	−0.366
CA.PGAv.1.6.scaffold26.14	*GPX7*	12	202945213	202961009	170	19,447.19	4.98	26.71	80.24	−0.356
CA.PGAv.1.6.scaffold559.5	*GPX8*	scaffold559.5	879793	886040	239	26,537.23	9.2	33.02	72.18	−0.252

## Data Availability

Data is contained within the article and Appendix A.

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
