# Peer review of "Characterization of GPX Gene Family in Pepper (Capsicum annuum L.) under Abiotic Stress and ABA Treatment"

_ijms, 2024, doi:10.3390/ijms25158343_

Round 1

Reviewer 1 Report

Comments and Suggestions for Authors

> Abstract the main objective is not clear, is it just characterization or want to identify the genes response against the stresses? Need more clear the objective part in abstract. 

> Keywords are hitting the title and abstract part need to reflects the title or abstract not the same. 

> Line 26 introduction is starting with very broad like plant level, my suggestion is to present your topic for your introduction. It's not review article.

> Line 86-88, statement for future recommendations is too general. If authors explain the objective in the sense of finding or results may be better for reader point of view. 

> Figure representation is really interesting in the reader point of view. But the figure legends are too brief. If authors explain the legends of the manuscript in detail may be more helpful to understand more about the figure and analysis. 

> In Figure 5 c UTR portion should remove from the Gene Structure which are the useless to explain the gene length. 

> Figure 8 colors with respect to words making confusion in understanding for the reader, its better to change the color of words or circle fills. 

> Figurer 10 expression analysis bars are very small which are hard to understanding. My suggestion is to change with line graph with respect to time points on x-axis may be better to explain them and compare them. 

> Material and Method section is clear and easy to understand. No comments to change it from my end. 

Author Response

Dear reviewer 

I sincerely thank you and the anonymous reviewer for your helpful comments and suggestions on our manuscript entitled “Characterization of GPX Gene Family in Pepper (Capsicum annuum L.) under Abiotic Stress and ABA Treatment”. I believe that our work has benefited substantially from your valuable input.

According to your comments and suggestions, we have revised the manuscript point by point. The corresponding changes in the manuscript have been highlighted, deleted words are not shown to avoid confusions and the responses to your comments are listed at the end of this letter point by point.

Again, We believe that the manuscript is now suitable for publication in IJMS.

Best wishes and kindest regards.

Yours sincerely.

>Comments and Suggestions for Authors

> Abstract the main objective is not clear, is it just characterization or want to identify the genes response against the stresses? Need more clear the objective part in abstract. 

Response: Thank you for your valuable comments. The original manuscript has been revised as follows, with the modified parts marked in red font:

Abstract: Plant glutathione peroxidases (GPXs) are important enzymes for removing reactive oxygen species in plant cells and are closely related to the stress resistance of plants. This study identified the GPX gene family members of pepper (Capsicum annuum L.) "CM333" at the whole genome level to clarify their expression patterns and enzyme activity changes under abiotic stress and ABA treatment. The results showed that 8 CaGPX genes were unevenly distributed over four chromosomes and one scaffold of the pepper genome, and their protein sequences had Cys residues typical of the plant GPX domains. Analysis of collinearity, phylogenetic tree, gene structure, and con-served motifs indicates that the CaGPX gene sequence is conserved, structurally similar, and more closely related to the sequence structure of Arabidopsis. Meanwhile, many cis-elements involved in stress, hormone, development and light response were found in the promoter region of CaGPX gene. In addition, CaGPX1/4 and CaGPX6 were basi-cally expressed in all tissues, and their expression levels were significantly upregulated under abiotic stress and ABA treatment. Subcellular localization showed that CaGPX1 and CaGPX4 are localized in chloroplasts. Additionally, the variations in glutathione peroxidase activ-ity (GSH-Px) mostly agree with the variations in gene expression. In summary, the CaGPXs gene may play an important role in the development of pepper and their response to abiotic stress and hormones.

> Keywords are hitting the title and abstract part need to reflects the title or abstract not the same. 

Response: Thank you for your valuable comments. The original manuscript has been revised as follows:

Keywords: pepper; glutathione peroxidase; abiotic stress; expression pattern

> Line 26 introduction is starting with very broad like plant level, my suggestion is to present your topic for your introduction. It's not review article.

Response: Thank you for your valuable comments.We have removed the background section introducing the plant level in the original manuscript and started with the introduction of glutathione antioxidant enzymes in the background section. The modified part is marked in red font in the original manuscript, as follows:

Glutathione peroxidase (GPX; EC 1.11.1.9) is not only an important member of the ascorbate-glutathione cycle (AsA-GSH) but also one of the important enzymes for clearing ROS in cells [1,2]. It belongs to the nonheme thiol peroxidase family [3], which can use glutathione (GSH) or thioredoxin reductase (TRX) as reducing agents to catalyze the reduction of hydrogen peroxide (H2O2), organic hydrogen peroxide oxides, and lipid peroxides into water or corresponding alcohols, thereby maintaining the balance of ROS in plants and protecting cells from the toxic effects of high concentrations of ROS [4,5]. People's understanding of GPX and research on its functional mechanism first started from animals, and it was discovered in 1957 by extracting mammalian red blood cells for enzyme testing and H2O2 reaction [6]. Since GPX in mammals uses GSH as an electron donor to reduce peroxides such as H2O2, the name GPX was derived from this [7]. The study of GPX in plants started late. Criqui et al. [8] isolated and identified the first plant-derived GPX from tobacco, and then successively found GPX homologous genes in different plants. In plants, GPX usually exist in the form of monomer [9], and almost all eukaryotic genomes contain coding genes of the GPX family, exhibiting high sequence similarity in conserved motifs and domains [10]. Compared with mammalian GPXs, plant GPXs contain cysteine (Cys) in their active sites, and when a large amount of peroxides accumulate in the plant, the three conserved Cys residues at the N-terminus will be converted into sulphenic acid, which forms an intramolecular disulfide bond with the separated conserved Cys residues [11,12]. The disulfide bond is subsequently reduced by the thioredoxin (Trx) or glutathione (Grx) system, while plant GPX is usually reduced by Trx [13], and most mammalian GPXs contain selenocysteine instead of Cys residues in their catalytic active sites, which can be reduced by Grx and Trx [14].

> Line 86-88, statement for future recommendations is too general. If authors explain the objective in the sense of finding or results may be better for reader point of view. 

Response: Thank you for your valuable comments. The original manuscript has been revised as follows:

In addition, we analyzed the physical and chemical properties, protein secondary/tertiary structures, chromosome localization, collinearity, phylogenetic relationship, gene structure, conserved motifs, cis-elements, protein interaction network, GO enrichment, and subcellular localization of the identified GPX family members. Furthermore, we conducted expression profiling and enzyme activity change analysis to understand their response to abiotic stress (i.e., cold and drought, salt) and ABA signaling.

> Figure representation is really interesting in the reader point of view. But the figure legends are too brief. If authors explain the legends of the manuscript in detail may be more helpful to understand more about the figure and analysis. 

Response: Thank you for your valuable comments. We have provided a more detailed description of the figure legends, and the modified parts are marked in red font in the original manuscript,such as:

Figure 1. Multiple sequence alignment of GPX protein sequences from pepper, rice and Arabidopsis. Sequence alignment was performed using the ClusterW method in MAGA11 and the alignment results were visualized with Jalview. The bottom two yellow column areas indicate the degree of base similarity, the black part indicates the common base, and the higher the column, the higher the base similarity. The three conserved domains found in most animal and plant GPXs are marked with black boxes; The amino acid residues that form part of the catalytic sites of GPXs are labeled with different colored triangles, including the strictly conserved Cys (C), Gln (Q), Asn (N), and Trp (W); Other conserved domains are marked with a red box.

Figure 3. Collinearity analysis of the GPX gene family between pepper and Arabidopsis. Gray lines denote the collinear blocks between pepper and Arabidopsis genomes and the lines of different colors represent collinear gene pairs of GPXs in pepper and Arabidopsis. Different colored squares 1-5 represent Arabidopsis chromosomes and their length, and different colored Ca-1-Ca-12 represents the pepper chromosome and its length. The outermost two circles represent gene density.

> In Figure 5 c UTR portion should remove from the Gene Structure which are the useless to explain the gene length. 

Response: Thank you for your valuable comments. We have removed the UTR part in Figure 5c and redrawn it, and made modifications to the description of this part in the original manuscript. The modified parts are marked in red font in the original manuscript.The above is the modified image, and the below is the original image.

> Figure 8 colors with respect to words making confusion in understanding for the reader, its better to change the color of words or circle fills. 

Response: Thank you for your valuable comments. We have modified the color and size of the font in Figure 8 and changed the filling colors of some circles to make the image look clearer.The left is the original image, and the right is the modified image.

> Figurer 10 expression analysis bars are very small which are hard to understanding. My suggestion is to change with line graph with respect to time points on x-axis may be better to explain them and compare them. 

Response: Thank you for your valuable comments. We have changed the X-axis in Figure 10 to the time points and redrawn the graph, modifying the description of this part in the original manuscript. The modified parts are marked in red font in the original manuscript.The above is the modified image, and the below is the original image.

> Material and Method section is clear and easy to understand. No comments to change it from my end. 

Reviewer 2 Report

Comments and Suggestions for Authors

As I indicate in the general evaluation of the work, its quality is high. The role of genes from the glutathione peroxidase family in pepper is evaluated in a very precise and exhaustive manner.  BUT, I need the authors to better describe the differences between their work and the work of Wang et al. (2023) published in the journal Gene. The work under review does not seem very novel, making this a key point for its final acceptance. A list of advantages and methodological improvements is welcome, but it is not sufficient for the authors to state that Wang's work does not address certain studies that are considered here, such as the relationship of these genes with the ABA response.

Authors should use the same terminology by consensus and not confuse words such as hydropathicity versus hydrophilicity.

The conclusion should synthesize and analyze the results, discuss their implications and relevance, acknowledge limitations, and suggest areas for future research. This provides a comprehensive and critical perspective on the study, beyond simply reiterating the results.

Author Response

Dear reviewer

I sincerely thank you and the anonymous reviewer for your helpful comments and

suggestions on our manuscript entitled “Characterization of GPX Gene Family in Pepper

(Capsicum annuum L.) under Abiotic Stress and ABA Treatment”. I believe that our work has

benefited substantially from your valuable input.

According to your comments and suggestions, we have revised the manuscript point by

point. The corresponding changes in the manuscript have been highlighted, deleted words are

not shown to avoid confusions and the responses to your comments are listed at the end of

this letter point by point.

Again, We believe that the manuscript is now suitable for publication in IJMS.

Best wishes and kindest regards.

Yours sincerely.

>Comments and Suggestions for Authors

>As I indicate in the general evaluation of the work, its quality is high. The role of genes from

the glutathione peroxidase family in pepper is evaluated in a very precise and exhaustive

manner. BUT, I need the authors to better describe the differences between their work and

the work of Wang et al. (2023) published in the journal Gene. The work under review does

not seem very novel, making this a key point for its final acceptance. A list of advantages and

methodological improvements is welcome, but it is not sufficient for the authors to state that

Wang's work does not address certain studies that are considered here, such as the relationship

of these genes with the ABA response.

Response: The analysis and identification of GPX family by wang et al. is a very good work, but

our research is more comprehensive and systematic, and there are some improvements in the

method. I will explain the differences between the work of wang et al. and ours from the following

aspects:

  1. Firstly, we use different databases, and the number of genes identified is different. Wang et al.

used the "Zunla" genome database and identified 5 CaGPX genes, while we used the "CM334"

genome database and identified 8 CaGPX genes.

  1. The five CaGPX genes identified by Wang et al. were distributed on chromosomes 1, 9 and 12

of pepper, while in our results, CaGPX genes were also distributed on chromosome 6 and scaffold,

among which the gene CaGPX4 on chromosome 6 was confirmed to play an important role in the

growth and development of pepper and resistance to stress in subsequent expression pattern

verification.

  1. In terms of collinearity, we not only analyzed the collinear gene pairs of the CaGPX gene in

pepper and the AtGPX gene in Arabidopsis, but also calculated their Ka/Ks values. The calculated

results are listed in Table S2. This also reflects the conservation of the pepper CaGPX gene from

an evolutionary perspective.

  1. In terms of phylogenetic tree, we first calculated the amino acid substitution models of GPX

proteins from 7 species, and the calculation results are included in the attached file 1. Then, weselected the LG+G model and used maximum likelihood (ML) method to construct the

phylogenetic tree. However, Wang et al. used the adjacency method.

  1. In terms of conserved motifs and gene structure, Wang et al. only analyzed 5 CaGPX genes,

while we added the model crops Arabidopsis and rice to analyze and compare the conserved

motifs and gene structures of the three species. In addition, as the first reviewer said, TUR does

not reflect gene length well, which was avoided in our study. This also makes our results more

accurate and indirectly reflects the conservation of the CaGPX gene.

  1. In terms of cis-acting elements, we have carried out statistics on the number and classification

of elements to make the results more clear. The analysis results of this part are also uploaded in

the additional materials.

  1. In addition, we also conducted GO enrichment and predicted the interactions between GPX

family proteins, which were not involved in Wang et al.'s study. Our results further demonstrate

that the CaGPX gene family plays a role in reactive oxygen species scavenging and antioxidant

defense, and also indicate that this family can work together with other enzyme family members to

exercise its functions.

  1. In terms of relative gene expression levels, our study included ABA treatment and conducted

experimental verification. There are studies indicating that some members of the GPX gene family

can be regulated through ABA-dependent signaling pathways. Therefore, it is necessary to study

the relative expression levels of GPX gene family members under ABA treatment. However, as

you mentioned, we cannot say that Wang's work did not involve certain studies considered here.

Therefore, we have made modifications to the content you mentioned in the discussion section.

  1. Secondly, we constructed Pac402-GFP-CaGPX1 and Pac402-GFP-CaGPX4 vectors to validate

the subcellular localization of CaGPX1 and CaGPX4, which was not addressed in Wang et al.'s

study.

  1. Finally, we conducted experimental verification on the enzyme activity of GPX under

different stress treatments and ABA treatments, and observed the phenotypic changes of pepper

plants at different time periods after treatment. This is also an auxiliary explanation for the relative

expression levels of genes and makes our experimental results more reliable.

The above are the improvements and enhancements in our research on the GPX gene family, as

well as the differences from the work of Wang et al.

>Authors should use the same terminology by consensus and not confuse words such as

hydropathicity versus hydrophilicity.

Response: Thank you for your kind reminder. The original manuscript uses hydropathicity

uniformly and is marked with red font.

>The conclusion should synthesize and analyze the results, discuss their implications and

relevance, acknowledge limitations, and suggest areas for future research. This provides a

comprehensive and critical perspective on the study, beyond simply reiterating the results.

Response: Thank you for your valuable comments. The original manuscript has been revised as

follows:In conclusion, a total of 8 members of the CaGPX gene family were identified in the

"CM334" genome database of pepper. Analysis of their physical and chemical properties, multiple

sequence alignment, phylogeny, collinearity, and gene structure revealed that the family had

conserved sequence and stable structure. Furthermore, GO annotation, protein interaction network,

subcellular localization, tissue-specific expression profiling, and expression profile analysis under

different osmotic stress and ABA treatment suggest that certain CaGPX genes play distinct roles in

pepper development and stress response and may contribute to the ABA signaling pathway.

However, the exact function of these genes remains to be elucidated in future studies. In addition,

these findings will lay the foundation for studying the role of the CaGPX gene in the development

of capsicum, as well as using CRISPR/Cas system overexpression, knockout, and other functional

verification schemes.
